# Bin2Vec: A Better Wafer Bin Map Coloring Scheme for Comprehensible Visualization and Effective Bad Wafer Classification

**Junhong Kim , Hyungseok Kim, Jaesun Park, Kyounghyun Mo and Pilsung Kang \***

School of Industrial Management Engineering, Korea University, Seoul 02841 , Korea;
junhongkim@korea.ac.kr (J.K.); hskim0263@korea.ac.kr (H.K.); jason_park@korea.ac.kr (J.P.);
momkdh2@korea.ac.kr (K.M.)
**\*** Correspondence: pilsung_kang@korea.ac.kr; Tel.: +82-2-3290-3383

**Abstract:** A wafer bin map (WBM), which is the result of an electrical die-sorting test, provides information on which bins failed what tests, and plays an important role in finding defective wafer patterns in semiconductor manufacturing. Current wafer inspection based on WBM has two problems: good/bad WBM classification is performed by engineers and the bin code coloring scheme does not reflect the relationship between bin codes. To solve these problems, we propose a neural network-based bin coloring method called Bin2Vec to make similar bin codes are represented by similar colors. We also build a convolutional neural network-based WBM classification model to reduce the variations in the decisions made by engineers with different expertise by learning the company-wide historical WBM classification results. Based on a real dataset with a total of 27,701 WBMs, our WBM classification model significantly outperformed benchmarked machine learning models. In addition, the visualization results of the proposed Bin2Vec method makes it easier to discover meaningful WBM patterns compared with the random RGB coloring scheme. We expect the proposed framework to improve both efficiencies by automating the bad wafer classification process and effectiveness by assigning similar bin codes and their corresponding colors on the WBM.

**Keywords:** wafer bin map (WBM); Bin2Vec; Word2Vec; bad wafer classification; convolution neural network

## 1. Introduction

The semiconductor market is growing at a rapid pace with the advent of the fourth industrial revolution, characterized by the widespread use of the Internet of Things and the use of artificial intelligence technologies in daily life [1,2]. To connect not only traditional devices such as PCs and smartphones but also other electronic devices such as automobiles and home appliances to networks, and to store and process the huge amounts of data obtained by these devices, the demand for various types of ultra-compact and highly integrated semiconductor products has significantly increased [1]. To meet these requirements, ultrafine process in semiconductor production technology has consistently evolved and has reached the physical limit with the help of advanced manufacturing processes.

The semiconductor manufacturing process consists of three main steps: (1) wafer fabrication; (2) wafer test, i.e., circuit measurement for defective die/chip identification; and (3) packaging, i.e., cutting the finished chips to a specific size to assemble them and make the final product such as DRAMs or solid-state drives [3]. Since hundreds of individual equipment are involved in the semiconductor manufacturing process, more than three months of production time is required. The overall yield of th semiconductor manufacturing industry is relatively lower than that of other manufacturing industries [4] because of the highly complex manufacturing processes involved; hence,

yield management is one of the key factors to gain a competitive advantage in the market. It is obvious that developing a new product faster than competitors is necessary for market leadership, but a company can also dominate the market when their mass production system is running with a high and stable yield. The company can take an advantage of price competitiveness through cost reduction, which results in increased market share and profits. Since the semiconductor manufacturing process has becoming more precise and highly integrated, it is increasingly difficult to secure high yields for various newly developed products [5].

The electrical die sorting (EDS) test is conducted to check the electrical operation state of each die on a wafer to identify defective dies and determine whether the current wafer can move on to the packaging stage [2]. From the perspective of yield management, it is very important to find the cause of defects in the semiconductor production process by examining the EDS test result [6]. EDS tests are conducted based on predefined conditions, which vary with the characteristics of products. Once a set of EDS tests is completed, a single decimal bin code is assigned to each die on the wafer. This bin code indicates a certain combination of tests that the die passed or failed. Theoretically, if there are 10 different tests, the possible number of bin codes is 1024 ($2^{10}$), which is represented as a decimal number ranging from 0 (passed all tests) to 1023 (failed all tests).

A wafer bin map (WBM) is an image that shows the relative location of each die and its corresponding bin code. WBMs are commonly used to help engineers understand the overall state of a wafer based on the EDS test results, as shown in Figure 1. The large circles in both images represent a single wafer while the small squares represent individual dies. The only difference between Figure 1a,b is the coloring scheme used for bin codes. All bin codes, except the code indicating that all EDS tests were passed, are presented as the same gray color in Figure 1a, whereas different RGB values are assigned different bin codes in Figure 1b.

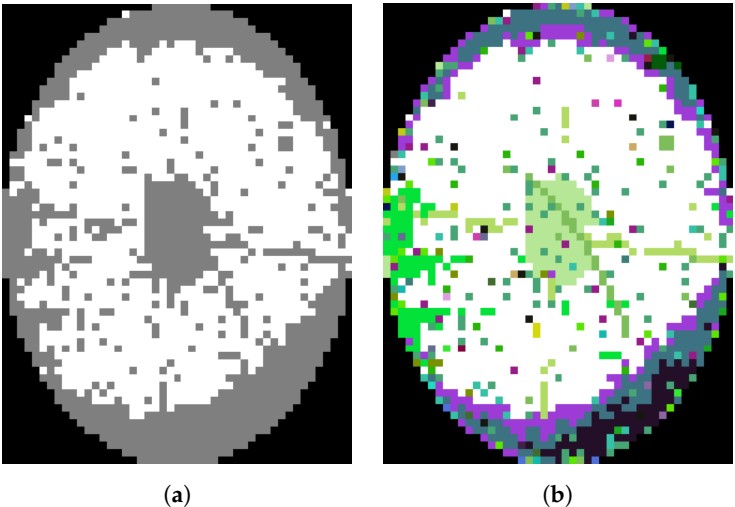

(**a**)                        (**b**)

**Figure 1.** Examples of WBMs with different coloring schemes: (**a**) WBM in which bins that passed all EDS tests are white and all the other bins are gray, (**b**) WBM in which bins that passed all EDS tests are white and other bins are randomly assigned a color according to the bin code.

Once the fabrication step is completed, various EDS tests are conducted for each wafer and the results are visualized by a WBM, based on which experienced engineers with sufficient domain knowledge determine whether the wafer should be delivered to the packaging step (normal/good wafers) or discarded (bad wafers) [7]. Since this decision is made solely on the basis of the experienced engineers' knowledge, different engineers may make different conclusions because of their biased domain expertise. In other words, the same wafer that is considered "normal" by one engineer may be deemed "bad" by another. More specifically, when a WBM is inspected by an engineer, he/she makes a bad wafer decision by focusing only on the types of test failures that are familiar based on

his/her experiences. In other words, he/she looks at only a few bin codes (WBM colors) to make a decision rather than considering all the bin codes. Consequently, depending on the WBM colors that the engineer focuses on, the same wafer can be identified as either "normal" or "bad." This problem is mainly caused by the limitations of the current WBM coloring scheme. When the gray WBM is used, significant information loss is inevitable since the engineer can only see the dies that failed at least one test; it is impossible to see which dies failed at what specific tests. With the random RGB coloring scheme, the role of color is to indicate unique bin codes; similar colors do not mean that the failed tests they represent are physically or electrically similar. Thus, existing WBM colors are not fully used since engineers use color information only to distinguish one bin code from another. If WBM colors can provide the relationship between two bin codes, i.e., two sets of failed tests, then engineers can better understand the WBM and identify significant patterns for improved decision-making.

To address the limitations of the abovementioned WBM coloring schemes, we propose a new, neural network-based WBM coloring scheme called bin-to-vector (Bin2Vec) that can preserve the relationship between different bin codes to help engineers better understand the WBM and help identify significant patterns on bad wafers. The Bin2Vec maps a scalar bin code onto a three-dimensional continuous vector in order to assign a unique set of RGB values to the bin code. The main idea behind Bin2Vec is that if two dies are physically close to each other, then it is highly likely that their EDS results are very similar; they might even pass and fail the same tests together. To realize this idea, Bin2Vec adopts the word-to-vector (Word2Vec) structure, which has been successfully used in various natural language processing tasks [8,9]. We expect that once the Bin2Vec succeeds in learning the local structure of EDS test results of closely located dies, the resulting RGB codes can not only discriminate one bin code from another but the RGB code can also represent the EDS test result similarity between any pair of bin codes. Moreover, these similarities can be visualized in a two-dimensional space to help engineers better understand different WBM patterns. In addition to the Bin2Vec, we also propose a convolutional neural network (CNN)-based bad wafer classification model, which has been done manually by experienced engineers. CNNs have shown excellent performance in various areas such as audio processing and image processing tasks, including image classification [10–15]. The bad wafer classification model takes the WBM as the input image, and determines whether it is normal or bad as the output of the network. To train the CNN model, historical data consisting of WBMs and their corresponding normal/bad labels determined by experienced engineers are used. If the CNN-based bad wafer classification model shows a certain level of accuracy, then WBMs can be automatically classified without the intervention of engineers.

The rest of this paper is organized as follows. In Section 2, we briefly review previous studies related to WBM-based bad wafer classification models. Section 3 discusses data collection and preprocessing, followed by a demonstration of the Word2Vec and proposed Bin2Vec methods. Section 4 describes the experimental design, including the proposed CNN model structure and benchmarked classification models. Experimental results are discussed in Section 5, and the limitations of the current study and future research directions are discussed in Section 6.

## 2. Literature Review

The mainstream research on WBM analysis has focused on the discovery of meaningful patterns or clusters from bad wafers. In this framework, bad wafers are manually filtered first, and then various analytical techniques are applied to find significant patterns or clusters based on their WBMs [16–20]. In the early 2000s, Huang et al. [16] found wafer clusters by applying a $3 \times 3$ median filter to the WBM to replace isolated chips with the median value of neighboring chips. Since median filtering has a blurring effect, it becomes difficult to discover defective wafer patterns with high precision. Instead of a median filter, Wang Wang [17] applied a mean filter to the WBM for the purpose of denoising, and found defective wafer clusters. As a clustering method, they employed spectral clustering based on fuzzy C-means (FCM) with kernel principal component analysis, given that FCM-based spectral clustering is known to be robust against outliers. Recently, a neural-network based method that

replaces the previous kernel principal analysis has also emerged to extract the principal singular triplet (PST) of a cross-correlation data, such as wafer images [21].

In the 2010s, Ooi et al. [18] proposed the segmentation with detection and small cluster removal (SDC) algorithm. The SDC algorithm automatically generated features for clustering WBMs by local yield conversion (LYC) method, which extracted the mean value of the "passed" and "failed" neighboring chips around the targeted chip. Through the cut-off filtering process, the LYC can extract the dominant wafer defect patterns. Taha et al. [19] proposed the Dominant Defective Patterns Finder (DDPFinder) algorithm that maintains the spatial dependence of defect patterns throughout the WBM, and considers the relative coverage of defect patterns during the clustering process. They randomly selected $k$ sample dies for cluster centroid initialization, which resulted in $k$-Voronoi regions in k-means clustering (KMC). The dominant defective patterns were extracted by the centroids of the Voronoi region to which defective wafers were assigned. They compared the performance of DDPFinder and eight benchmarked models (i.e., simplified subspaced regression network, randomized general regression neural network, support vector machines, and four artificial neural networks (ANNs, i.e., general regression neural, radial basis function, probabilistic neural network, and multi-layer perceptron)). The experimental results showed that DDPfinder outperformed the benchmarked methods with an accuracy of 99.78% based on a predefined defect wafer dataset. Liu and Chien [20] proposed a framework that removed random noise to find robust defective WBM pattern clusters based on a cellular neural network. To do so, they first used a moment-invariant method, which is one of the shape recognition techniques widely used in the field of image processing, to extract rotation and size-invariant features. Based on the extracted features, they applied a neural network-based adaptive resonance theory (ART) to find WBM clusters. Their method showed 97% clustering purity even with high defective rate in the highly noisy environment. The experimental results also showed that the proposed method resulted in higher purity, diversity, specificity, and efficiency than the four existing clustering methods (Ward, KMC, self-organizing map (SOM), and spectral). However, the main disadvantage of this method is that it cannot find combined defective patterns.

In addition to the task of finding significant defective WBM patterns, some studies have attempted to automate the process of finding defective patterns and assigning the WBMs of newly processed wafers to one of these patterns, which is formulated as a multiclass classification problem. Wang et al. [5] proposed a hybrid clustering method to define the characteristics of WBM clusters and assign individual WBMs to one of these discovered patterns. They first applied a spatial filter to WBMs to extract relevant features for defective pattern identification. Then, they used hybrid clustering to define cluster types based on the extracted features. To classify individual WBMs into one of these identified clusters (e.g., linear type, elliptical type, and ring type), various classification methods such as classification module, estimation module, Gaussian EM algorithm, and spherical shell algorithm, were employed. Li and Huang [22] proposed a hybrid SOM–SVM method to recognize and classify WBM defect patterns. In their framework, SOM was trained to discover representative WBM defect patterns, whose clustering membership became the target class of the SVM classifier. Experimental results showed that the proposed method can correctly classify the discovered defect patterns with more than 90% accuracy. Cheng et al. [4] attempted to extract significant features using polar Fourier transform and rotational moment invariants, both of which are commonly used in signal processing. They trained classifiers to classify six predefined WBM defect patterns, i.e., bulls-eye, blob, hat, ring, line, and edge, based on the extracted features. A decision tree, Naive Baye's classifier, bagging decision tree, and boosted logistic regression were employed as classification algorithms. Although this method showed good performance for simulated WBM datasets (68.33% accuracy), its performance deteriorated when applied to a real WBM dataset (52.58% accuracy). Another notable limitation is that the WBM defect patterns were predefined and were not changed over time. Since currently non-existent defective patterns can be observed during the actual manufacturing process, this method is not flexible enough to adapt to unexpected changes in manufacturing environment.

Liao et al. [23] proposed a defect-type detection method based on similarity searching to address problems of not only inflexibility caused by predefined defect patterns but also the difficulty of capturing complex defect patterns when more than two defect patterns occur simultaneously. To do so, they first performed a morphological sample generation process to reproduce various pattern derivations (shift, erosion, dilation, opening, and closing) from the original defect patterns. Then, they used two types of training datasets: predefined defect patterns determined by experienced engineers and dissimilar samples determined by the 1-SVM. The SVM classifiers were trained to classify the five defect patterns. The experimental results showed that the proposed method achieved 95% catching rate with only 5% false-alarm rate. However, this model can only be applied to predefined defect patterns. Adly et al. [24] and Adly et al. [25] employed ANN-based classifiers called simplified sub-spaced regression network and randomized general regression network to improve the WBM defect pattern classification performance. Their first proposed model [24] yielded superior and robust performance, in addition to improved computational efficiency compared with the benchmarked six methods. In the following study [25], they proposed a classification model with a bagging ensemble scheme that guaranteed relatively stable prediction accuracy and low variance. The proposed method achieved approximately 99.79% classification accuracy. Wu et al. [26] proposed a two-phase method that first discovered defect patterns and then classified WBMs into one of these patterns based on different image feature extraction methods. They employed random-based features, geometry-based features, and Hough transformation features for feature extraction. Then, the SVM classifier was used to classify the WBM. Their proposed model yielded 94.63% classification accuracy in nine-class (eight defect classes + no class) classification.

The aforementioned studies mainly focused on discovering representative defect patterns that appear on WBMs and classifying individual WBMs into one of these discovered classes. There are two common assumptions behind these studies: (1) defective WBMs can be easily classified with little effort, and (2) the classification result is highly accurate. However, these two assumptions might not be supported in actual semiconductor manufacturing processes. First, defective WBM classification is still manually conducted by experienced engineers, which is both time-consuming and labor-intensive. Second, some wafers determined as normal by one engineer are determined as defective by another engineer since engineers with different expertise investigate different bin codes to make their decisions. In other words, human judgment of defective WBM is not based on all the information embedded in the WBM but also on only a part of it. The information on WBM that is used solely depends on who investigates it. The second problem is closely related to the current WBM coloring scheme; RGB codes are randomly assigned to different bin codes. Hence, the colors in existing WBM are simply identifiers of bin codes and cannot preserve the physical or semantic relationship between them. If the relationship between bin codes can be understood through WBM colors, then contradictory decisions by engineers regarding defective WBMs can be reduced, making it possible to build an accurate autonomous WBM classification model that can classify every WBM in real time without any intervention by human experts.

## 3. Bin2Vec: A Convolutional Neural Network-Based WBM Coloring Scheme

### 3.1. Data Description

In this study, we used a total of 27,071 WBMs from nine product groups (DRAM, SSD, etc.) produced by a semiconductor manufacturing company in South Korea. The total number of bin codes available for each die is 10,002, as listed in Table 1. If a die does not pass a certain set of EDS tests, then one code between 1 and 9999 is assigned. The values 0, $-1$, and $-2$ refer to passing all the EDS tests, untested dies, and non-wafer area, respectively. Figure 2 shows an example of a WBM with random RGB coloring scheme (a), and a simplified binarization scheme (b) in which bin codes from 1 to 9999 are represented by the same color (blue in this example).

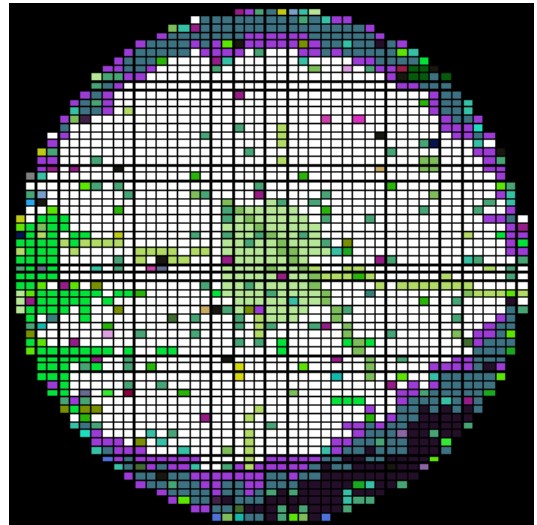
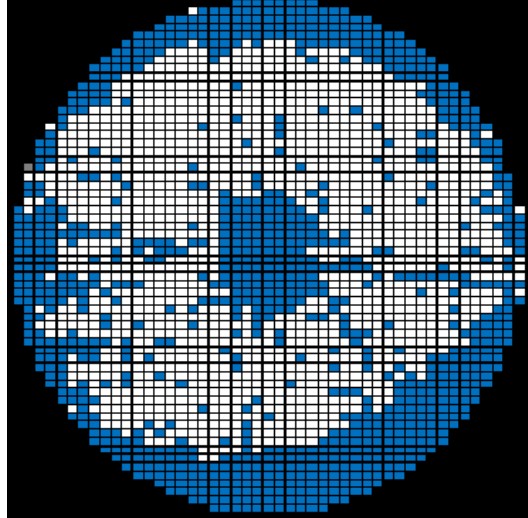

(**a**) Visualized by random RGB method    (**b**) Visualized by simplified method

**Figure 2.** Example of WBM.

**Table 1.** Values that can be assigned to WBM image pixels.

| Value | Description |
| --- | --- |
| $-2$ | Non-wafer part |
| $-1$ | Untested portion |
| 0 | Not all results from 1 to 9999 |
| 1~9999 | Code indicating a set of EDS tests that the die does not pass |

*3.2. Word2Vec and Bin2Vec*

3.2.1. Word2Vec

Word2Vec is the most widely used neural network-based distributed representation method that represents a word by a numerical vector in a fixed dimension [27,28]. Traditional one-hot encoding cannot preserve the semantic relationship because a word is represented by $|V|$-dimensional vector ($|V|$ is the number of unique words in the dataset) with only one element with the value 1, and all other elements being 0. In contrast, Word2Vec can preserve the semantic relationship between words.

There are two Word2Vec structures: continuous bag-of-words (CBOW) and Skip-Gram that have the exactly reversed structures. CBOW takes the neighborhood words as the input of the network and is trained to correctly predict the targeted word, whereas Skip-Gram takes the targeted word as the input of the network and is trained to correctly predict the neighborhood words. We adopted the Skip-Gram structure, as shown in Figure 3 [27], since it is known to be more effective in learning infrequent words. The bin codes in this study are equally important regardless of the frequency of their appearance. The Skip-Gram model consists of the input, projection, and output layers. Assume that we are attempting to find the $d-$dimensional word vectors for a total of $|V|$ words in the dataset ($d << |V|$), and the window size (the number of neighborhood words to predict in either direction from the targeted word) is set to $c$. The input layer takes the one-hot encoding vector of the targeted word, whereas the output layer consists of the one-hot encoding vectors of the words surrounding the targeted words. The projection layer consists of $d$ hidden nodes such that the hidden weight matrix **W** between the input layer and projection layer becomes $|V| \times d$, and the hidden weight matrix **W**$'$ between the projection layer and output layer becomes $d \times |V|$. After the training process is completed, either the row vector of **W** or the column vector of **W**$'$ can be used as word vectors.

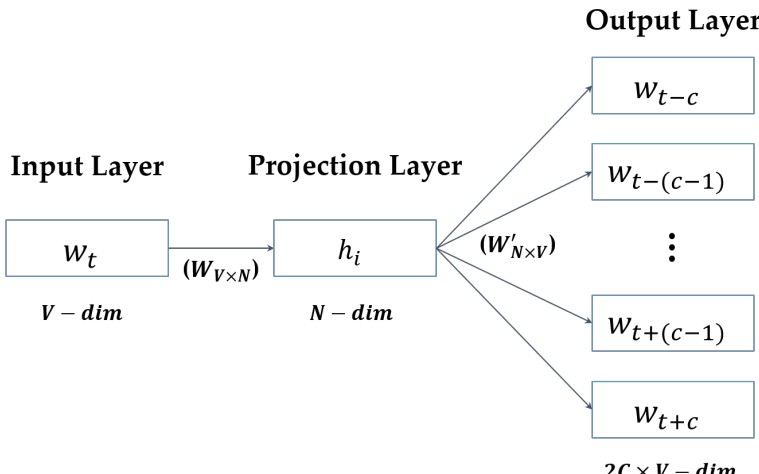

**Figure 3.** The Skip-Gram model architecture.

The purpose of the Skip-Gram model is to maximize the probability of generating the actual surrounding words for a given target word. More specifically, given a sequence of training words $w_1$, $w_2$, $w_3$, $\cdots$, $w_T$, the objective function of the Skip-Gram model is to maximize the average log probability as shown in Equation (1):

$$\max \quad \frac{1}{T} \sum_{t=1}^{T} \sum_{-c \leq j \leq c, j \neq 0} \log p(w_{t+j}|w_t). \tag{1}$$

The probability of generating the surrounding (output) words given the target (context) word is computed using the softmax function as shown in Equation (2):

$$p(o|c) = \frac{\exp(u_0^T v_c)}{\sum_{w=1}^{W} \exp(u_0^T v_c)}, \tag{2}$$

where $u$ is the column vector of the hidden weight matrix $\mathbf{W}'$, and $v$ is the column vector of the hidden weight matrix $\mathbf{W}$. The Skip-Gram model learns the weight matrices $\mathbf{W}$ and $\mathbf{W}'$ by stochastic gradient descent optimization method. There are two practical drawbacks of Equation (2). First, it is burdensome to compute the denominator because all the words must be used to compute the softmax function. Second, frequent words such as "the" or "it" are over-trained, whereas infrequent words cannot be trained sufficiently. To resolve these issues, the negative sampling technique [29] is commonly used. In the negative sampling technique, only a few non-surrounding words are sampled to compute the denominator, and the sampling probability of each word is adjusted based on its frequency in the corpus: the probability of sampling frequently appearing words decreases, whereas the probability of sampling of rarely appearing words increases. Noise contrastive estimation [30] is an alternative to reduce the computational complexity of the Word2Vec model.

### 3.2.2. Bin2Vec

In this study, the original WBMs have the size $301 \times 301$ pixels. Each pixel in a WBM can have a value between $-2$ and 9999. Although the nine product groups have the same raw WBM size, the actual number of dies on the wafer along the horizontal and vertical axes are different. Table 2 lists the number of unique bin codes, the number of dies along the horizontal/vertical axis, as well as the number of total, good, and bad wafers and their proportions. Group 1 has the largest number of WBMs, containing 5464 WBMs with $23 \times 27$ dies. In contrast, Group 9 has the smallest number of WBMs, containing 1715 WBMs with $47 \times 51$ dies. For the current manufacturing process of the company, a total of 10,002 bin codes are possible for the dies, as explained with Table 1. However,

only a small portion of them actually appeared: Group 1 has the largest number of unique bin codes (401), whereas only eight bin codes appear for Group 6. Although the size of raw WBMs is $301 \times 301$ pixels, each die is represented by $3 \times 3$, $4 \times 3$, or $4 \times 4$ pixels according to the product type, and the same bin codes are assigned to the pixels for the same die. Hence, we resized the original WBM by aggregating the pixels for the same die so that a die is presented by a $1 \times 1$ pixel in the reduced WBM as shown in Figure 4.

**Table 2.** Raw data information of WBM.

| Group | No. Wafers | No. Good | No. Bad | Good (%) | Bad (%) | No. Bins | No. Dies (Horizontal) | No. Chips (Vertical) |
|---|---|---|---|---|---|---|---|---|
| 1 | 5464 | 4156 | 1308 | 76.06 | 23.94 | 401 | 23 | 27 |
| 2 | 5449 | 4771 | 678 | 87.56 | 12.44 | 77 | 29 | 50 |
| 3 | 3261 | 2621 | 640 | 80.37 | 19.63 | 43 | 41 | 36 |
| 4 | 2781 | 2578 | 203 | 92.70 | 7.30 | 108 | 47 | 62 |
| 5 | 2543 | 2145 | 398 | 84.35 | 15.65 | 371 | 29 | 26 |
| 6 | 2101 | 850 | 1251 | 40.46 | 59.54 | 8 | 32 | 38 |
| 7 | 1958 | 1403 | 555 | 71.65 | 28.35 | 82 | 33 | 58 |
| 8 | 1799 | 1535 | 264 | 85.33 | 14.67 | 206 | 26 | 26 |
| 9 | 1715 | 1542 | 173 | 89.91 | 10.09 | 108 | 47 | 51 |

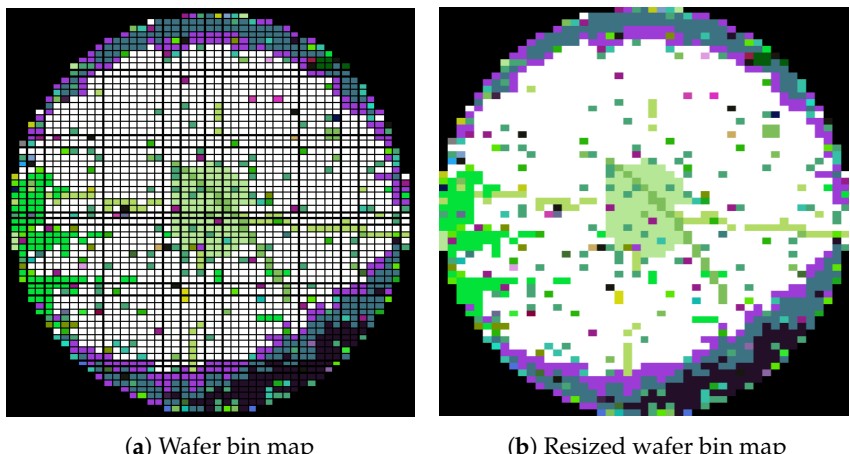

(**a**) Wafer bin map        (**b**) Resized wafer bin map

**Figure 4.** Example of wafer bin map and resized image from wafer chip size.

Currently, WBMs are colorized by the random RGB scheme in which a randomly selected RGB code is assigned to each bin code so that different bin codes can be distinguished on the WBM to help engineers identify meaningful bin patterns. The main drawback of this coloring scheme is that the relationships between bin codes are not preserved. In practice, some bin codes are closely related while others are completely unrelated. This relationship information is completely ignored by the current WBM coloring scheme. Encouraged by process engineers' domain knowledge that nearby dies often fail the same test together, we propose a new WBM coloring scheme called Bin2Vec by considering the location information of dies on the wafer. The proposed Bin2Vec transforms a one-dimensional scalar bin code into a three-dimensional embedding vector by employing the learning mechanism of the Word2Vec embedding model. Once the bin code is converted into a three-dimensional vector, it can be used not only to assign an RGB code, which can also be considered a three-dimensional vector, but also to compute the similarity between any two pairs of bin codes. For engineers who actually investigate WBM patterns, Bin2Vec can also provide the following information compared with random RGB coloring, i.e., the more similar the color, the more frequently the bin codes appear together in a local area on the wafer.

To train the Bin2Vec model, a total of 25 bin codes with size $5 \times 5$ are taken to make a training instance, as shown in Figure 5a. This square matrix is then reshaped into a 25-dimensional vector: the first column becomes the first five elements and the last column becomes the last five elements, as shown in Figure 5b. This sampling and reshaping process for a single wafer is repeated by sliding the

step size by one. For example, Group 1 products have 23 × 27 dies on a wafer, and we assign the index from top to bottom, and from left to right. The first training instance is made by taking 25 dies with index of 1–5 (horizontal) and 1–5 (vertical). If horizontal sliding is done first, then the second training instance is made by taking 25 dies with index of 2–6 (horizontal) and 1–5 (vertical). This process is repeated until the 25 dies with index 19–23 (horizontal) and 23–27 (vertical) are taken. With these vectorized bin codes, we employed the Skip-Gram structure [27] to learn the embedding vectors. The bin codes in the middle, i.e., the bin code in the third row and third column in the 5 × 5 bin matrix, or the 13th bin code in the 25-dimensional vector, were used as the input while the other 24 bin codes were used as the target of the Skip-Gram model, as shown in the first two steps in Figure 6. During the Skip-Gram model training, we excluded the following two cases because they do not have any significant information: (1) instances when all elements have the same bin code, and (2) instances that contain only "-2" (non-wafer part) and "0" (the die passed all the EDS tests).

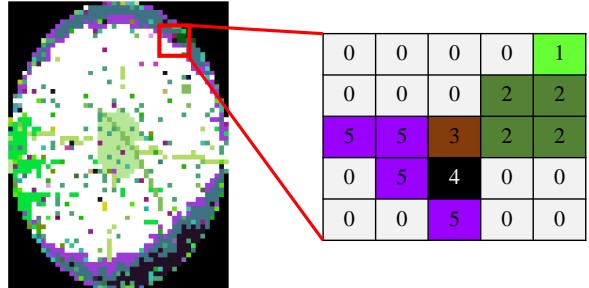

(**a**) Example of 5 × 5 window

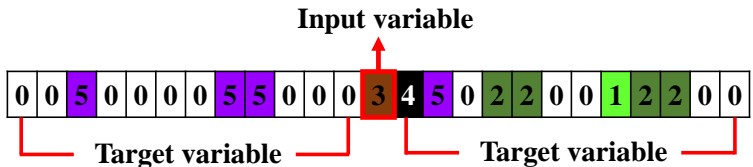

(**b**) Bin2Vec input/target variable

**Figure 5.** Data converted from WBM to Bin2Vec input structure.

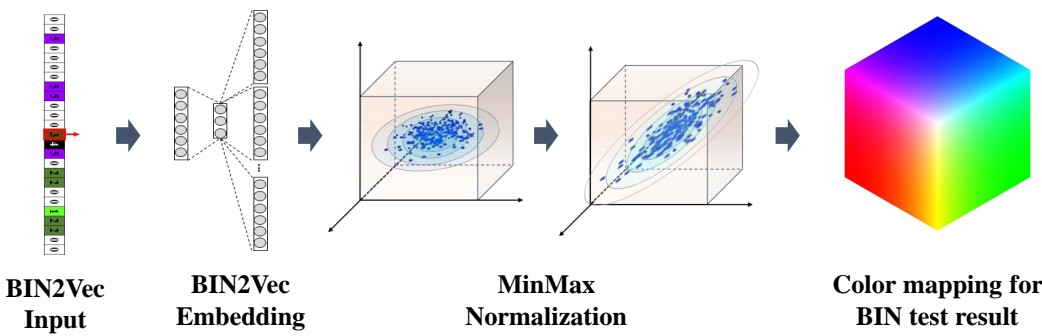

**Figure 6.** Bin code RGB color mapping framework based on Bin2Vec.

For the Bin2Vec model, the input vector dimension ($|V|$) is set to 401, which is the number of total bin codes that appeared at least once in the WBM dataset. The number of hidden nodes is set to 3 to assign the RGB code to the embedded bin codes. In an Natural language processing task, it is common to use a negative sampling technique or noise-contrastive estimation [30] for Word2Vec training to reduce the number of words used for the denominator of the softmax function, as shown in Equation (2). Without these techniques, Word2Vec would consider more than 100,000 words to compute the softmax function, causing a heavy computational burden. On the other hand, in the

Bin2Vec, we used all bins to compute the softmax function because there were only 401 distinctive bin codes. Once the Bin2Vec model training is completed, the range of three embedded dimensions can be different. To fully use the RGB code coverage, the embedded bin vectors are min-max scaled for all dimensions, as shown in Figure 6. Accordingly, each bin code has a value between 0 and 1 in the R-, G-, and B-axis, respectively.

Figure 7a is an example of a WBM with binary colorization scheme. The white color indicates that the dies in those areas passed all the EDS tests while the gray color indicates that the dies in those areas failed at least one of the EDS tests. The only information delivered to process engineers by the binary colorization scheme is whether each die passed all the tests or not. The failed tests cannot be distinguished because these are presented as the same gray color. To provide both types of information, the random RGB coloring scheme assigns different colors to different bin codes, as shown in Figure 7b. Although different bin codes are colored differently, it is not possible to determine which bins are semantically similar to others; the colors only serve as identifiers. Bin2Vec can provide all three types of information: (1) whether each die passed all the tests, (2) what die failed which tests, and (3) which bin codes are semantically related. Figure 7c shows the same WBM with the Bin2Vec coloring scheme. Compared with random RGB coloring, bin patterns are more noticeable with the Bin2Vec. The areas A, B, and C have similar green colors in Figure 8a, whereas black, purple, and brown are assigned to the dies in areas A, B, and C by the Bin2Vec, respectively (Figure 8b). In addition, Figure 8b shows that the bin codes along the edge are semantically similar since they are represented in yellow. On the other hand, they could also be misinterpreted as being very different since purple and green were assigned to the same area by the random RGB coloring scheme.

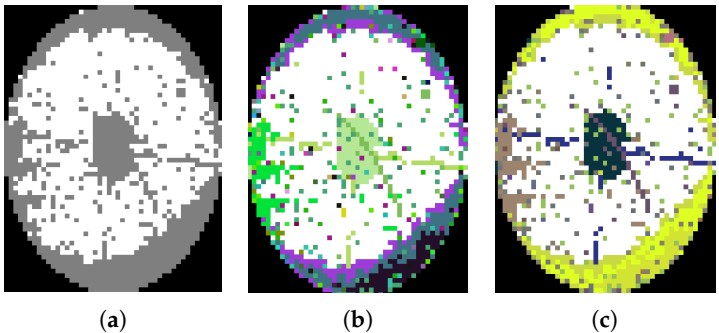

(**a**)  (**b**)  (**c**)

**Figure 7.** Various visualization methods in WBM: (**a**) Binarization, (**b**) Random RGB, (**c**) Bin2Vec.

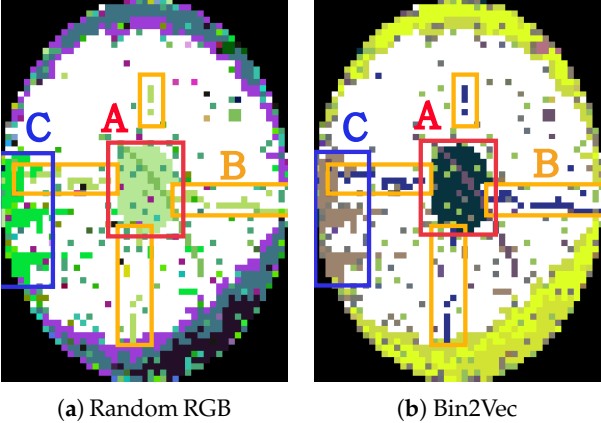

(**a**) Random RGB  (**b**) Bin2Vec

**Figure 8.** Bin patterns of random RGB and Bin2Vec.

Figure 9 shows two examples of WBMs where a significant bin pattern is recognized only with Bin2Vec. Figure 9a–c are the WBMs for a wafer and Figure 9d–f are the WBMs for another wafer. In Figure 9a,b, the bins seem to be randomly distributed and no significant patterns are recognizable.

However, it can be observed that there is a clear cross bin pattern in dark blue when Bin2Vec is applied. Similarly, it is hard to find a circular bin pattern in Figure 9d,e, but a circular bin pattern was discovered (color blue) by Bin2Vec.

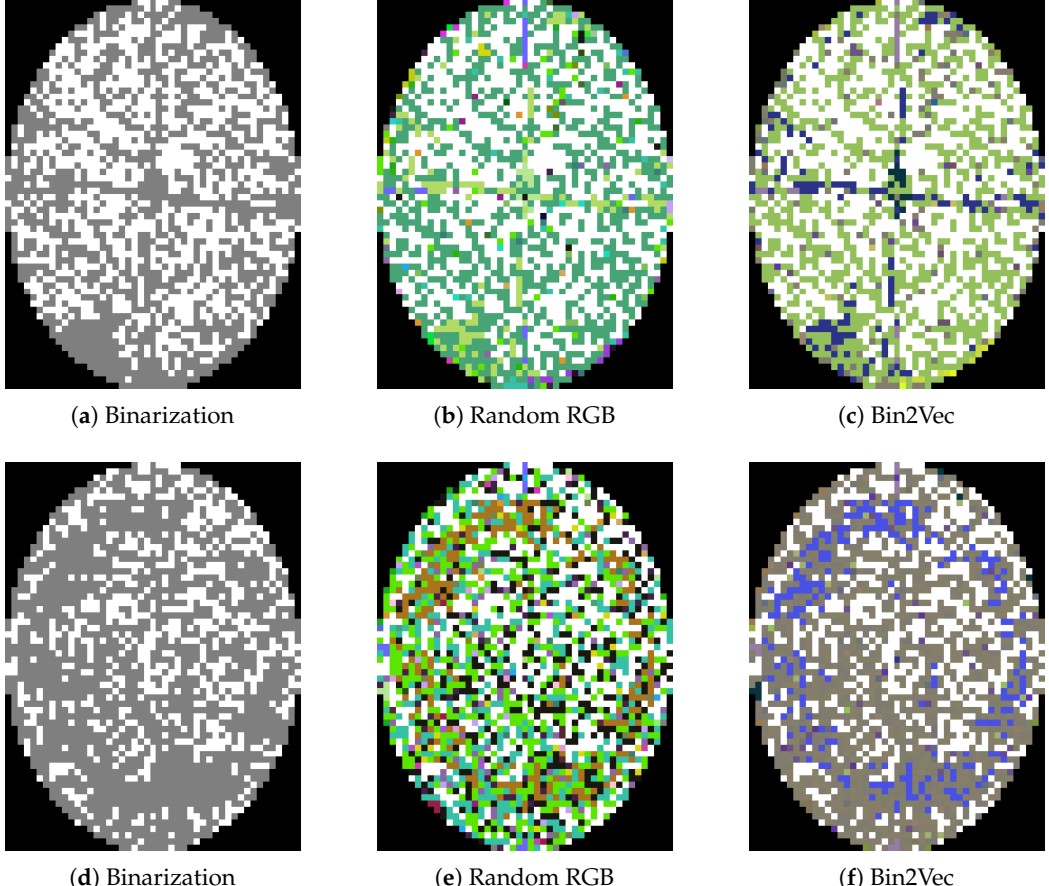

**Figure 9.** Example of WBM with obvious patterns when visualized by Bin2Vec methodology (Binarization, Random RGB, Bin2Vec).

Figure 10 shows other examples where small but significant bin patterns were discovered by Bin2Vec. In Figure 10a, only two dies passed all the EDS tests. Based on the random RGB scheme, it seems that many different bin codes are randomly distributed on the wafer. However, these bin codes are actually very closely related to each other such that their colors resulted in similar RGB codes with the Bin2Vec, as shown in Figure 10c. In Figure 10d, all dies failed at least one of the EDS tests such that the all areas on the wafer are color gray. With the random RGB scheme, it appears that there are three primary bin codes which are very different from each other. However, two of them are actually closely related to each other semantically (green and gray), and the other bin code (dark blue) is significantly different, which was discovered by the Bin2Vec.

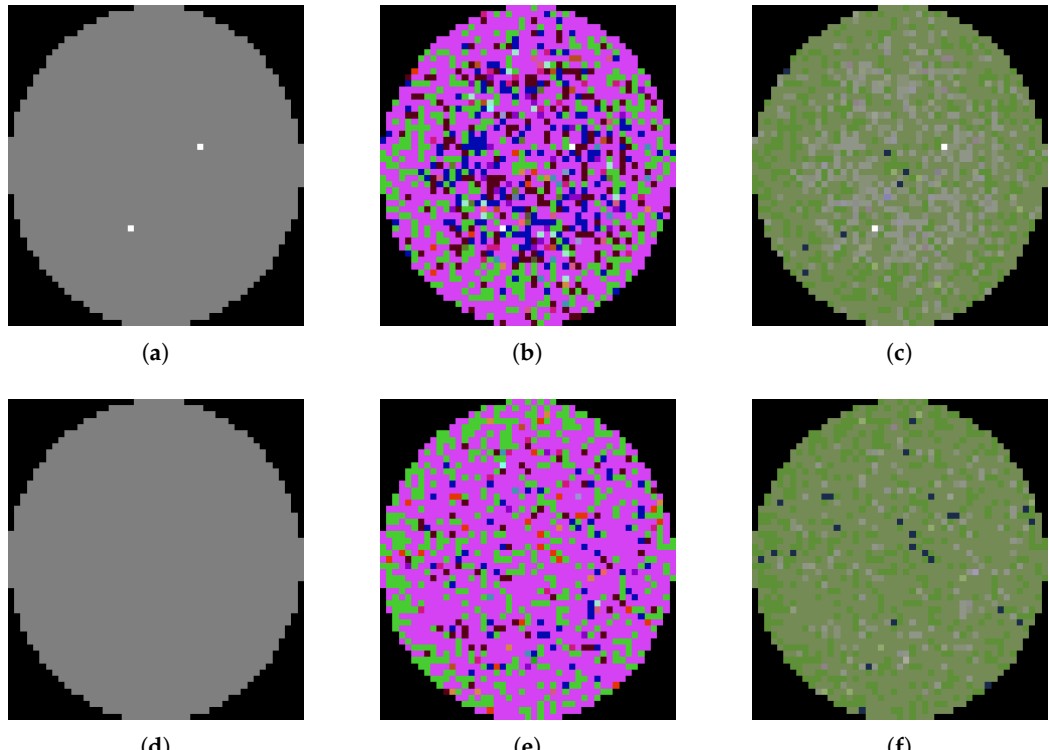

**Figure 10.** Examples of visualized novelty bin codes using Bin2Vec methodology (Binarization, Random RGB, Bin2Vec).

## 4. Experiment

In the semiconductor manufacturing process, WBMs are used as a source in faulty wafer identification. If a WBM is deemed good, then the corresponding wafer is delivered to the packaging stage. If it is deemed bad, then it is either re-processed or discarded. To identify bad wafers, we constructed a binary classification model: the input is the WBM of each coloring scheme, and the target (class label) is the good/bad identification results determined by expert engineers. An independent classification model was trained for each of the nine product groups. Since WBM can be considered as image data, we propose a CNN-based faulty wafer classification model. As benchmark algorithms, multilayer perceptron (MLP) with one hidden layer and random forests (RF) were employed. The training and test datasets were randomly partitioned by 8:2 for each product group, and the experiment was repeated 30 times for performance comparison.

### 4.1. Convolution Neural Network

#### 4.1.1. Convolution and Pooling Operation

In this study, CNNs extract useful features for classification from the image through the convolution operation. In image classification, 2$d$ convolution is commonly used. Let the feature map of $l$-th layer be $\mathbf{X}^l \in \mathbb{R}^{W \times H \times C}$, in which $W$ and $H$ denote the width and height of WBM, respectively; and $C$ denotes the channels (one for binarization, three for random RGB and Bin2Vec). The convolution operation is computed as shown in Equation (3):

$$\mathbf{X}^{l+1}_{w,h,c} = \sum_{c'=1}^{M} \sum_{m=1}^{M} \sum_{n=1}^{N} k_{c,m,n,c'} \mathbf{X}^{l}_{(w+m-\frac{M+1}{2}),(h+n-\frac{N+1}{2}),c''} \tag{3}$$

where $\mathbf{X}^{l+1}$ is the output tensor of the convolution operation; $w$, $h$, and $c$ denote the index of the width, height, and channel of the output tensor, respectively; and $k_{c,m,n,c'}$ is the weight used to conduct

the convolution with the width, height, and channel size of $m$, $n$, and $c'$, respectively. Figure 11a shows an illustration of convolution operation.

Once the convolution operation is completed, the pooling operation is conducted to reduce the width and height of features maps but preserve significant information, which helps improve the computational efficiency of CNN models. Max pooling and average pooling are the most commonly used pooling methods; we adopted max pooling since it focuses on the most important pixel in the receptive field. Figure 11b shows an example of max pooling operation with size of $2 \times 2$.

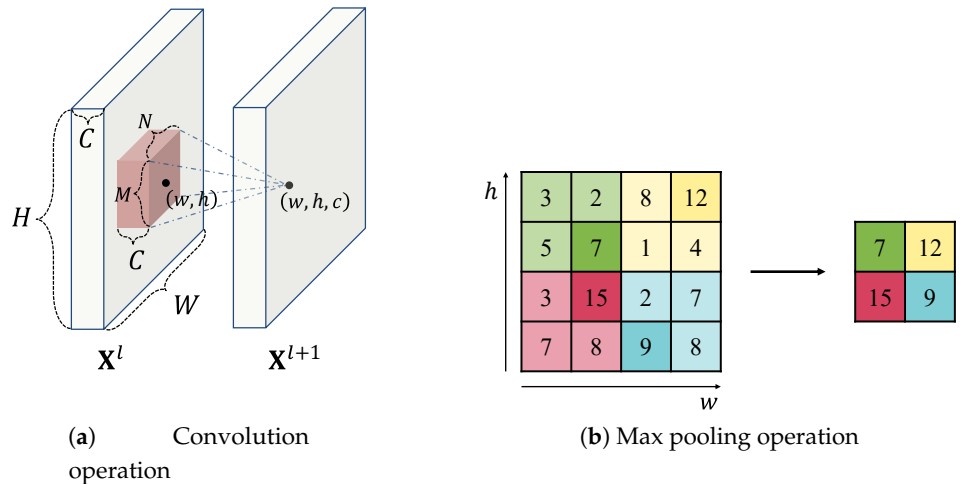

(**a**)    Convolution
operation

(**b**) Max pooling operation

**Figure 11.** Convolution and max pooling operations in CNN.

### 4.1.2. Convolutional Neural Network Architecture

A CNN is a special case of ANNs, as shown in Figure 12. CNN has been proven excellent performance in recent various image processing tasks [31,32]. Some representative CNN architectures include AlexNet [10], VGGNet [11], Inception [12], ResNet [13], and DenseNet [14].

The basic building blocks of a CNN are a convolution layer and pooling layer. Contrary to the basic ANN where the previous layer is fully connected to the next layer, a small region of the previous layer ($m$ by $n$ by $c'$), known as the receptive field, is connected to the next layer by the convolution operation in CNN. This convolution operation is repeated starting from the top-most left receptive field to the right-most bottom receptive field with the same convolution weights. Once the convolution operation is completed, another convolution operation with different weights is performed in the same manner. The number of different sets of convolution weights is known as the number of convolution filters between two layers. Since the same weights are used for a specified convolution filter, the number of weights to be learned is much smaller with the CNN than fully connected ANNs. In addition, a single convolution filter can find a significant local pattern over all regions in the previous layer using the same weights for the corresponding convolution filter.

Once the convolution step is completed, the size of the resulting tensor is reduced by pooling operation to improve the learning process efficiency. The pooling result depends on the size and method of pooling: the former decides how a large area is summarized while the latter decides how to represent the targeted area. Max pooling and average pooling are two common pooling methods: the former uses the largest value while the latter uses the average values of the targeted area.

After the set of convolution and pooling layers are processed, many commonly used CNN architectures place fully connected layers between the end of convolution building blocks and the output layer. For a classification task, the number of nodes in the output layer is set to the number of classes, and the output values of the CNN are computed using the softmax function to ensure that every output node value is between 0 and 1, and the sum of all output node values is 1. By doing so, the output node value can be interpreted as the probability of the input image belonging to the corresponding class. Because the number of weights to be learned is still quite large, some

practical techniques are used to improve training efficiency [33], such as adopting a rectified linear unit (ReLU) [10] as an activation function and batch normalization [34] during the weight learning by the gradient descent algorithm.

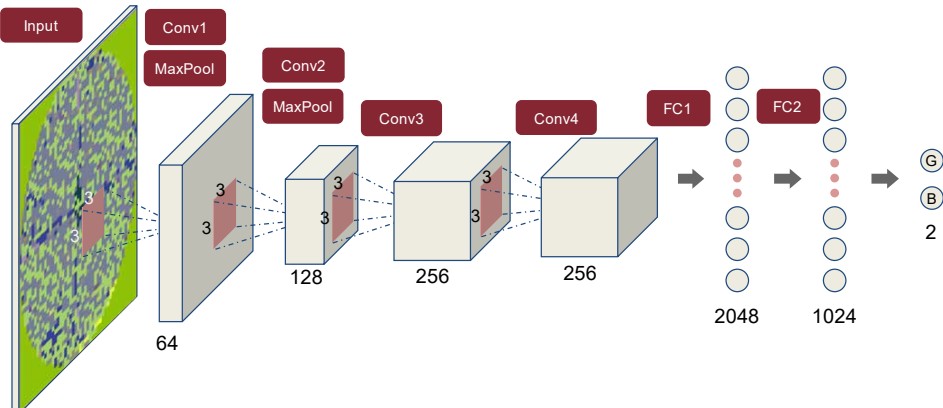

**Figure 12.** Convolutional neural network architecture used in this study.

The most commonly used input image size for CNN architecture is $224 \times 224$, which is the input size of the dataset provided by ImageNet [35], one of the largest labeled datasets for image processing. The maximum input image size in this study is $47 \times 62$. Hence, we used a simpler CNN architecture, i.e., the number of convolution and pooling layers is less than that of other well-known CNN architectures. Figure 12 shows the CNN architecture used in our study. Four convolution and pooling layers were used followed by two fully connected layers. For all convolution layers, $3 \times 3$ convolution operation with stride size of 1 was performed, followed by $2 \times 2$ max pooling with stride size of 2. The number of convolution filters are 64, 128, 256, and 256 for the first, second, third, and fourth convolution layers, respectively. The number of nodes used for the first and second fully connected layers are 1048 and 1024, respectively. The number of output nodes was set to two: good and bad.

We used ReLU and batch normalization after all of the convolution layers. The batchsize was set to 32. Given the imbalanced class distribution, i.e., the number of good wafers is generally much greater than that of the bad wafers, we constructed the mini-batch training set by sampling 16 good and 16 bad WBMs from the original training set. The learning rate started at $10^{-3}$, decayed by $10^{-1}$ every 5000 iterations after first 10,000 iterations, and trained 30,000 iterations in total. The first 20,000 iterations were trained with Adam [36] optimization, followed by 10,000 with mini-batch gradient descent (MGD).

### 4.2. Multilayer Perceptron

A feedforward MLP with three hidden layers was employed in this study. As shown in Figure 13, the number of hidden nodes was set to 1024, 1024, and 512 for the first, second, and third hidden layers, respectively. The ReLU activation function was used for the first and second hidden layer while the softmax function was used for the third hidden layer. Batch normalization is applied to each layer and the dropout rate of 0.5 was used [37] for the weights between the third hidden layer and output layer. The learning rate started at $10^{-3}$, decayed by $10^{-1}$ every 5000 iterations, and trained 15,000 iterations in total. We used the Adam optimizer for the first 10,000 iterations and then changed it to MGD. Similar to CNN, the batch size was set to 32, consisting of 16 good and 16 bad WBMs.

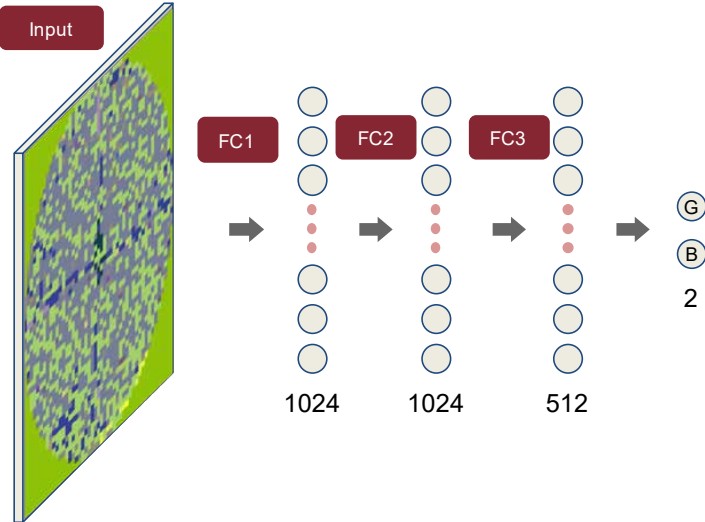

**Figure 13.** Multilayer perceptron architecture used in this study.

### 4.3. Random Forest

RF, which is a special case of the decision tree-based ensemble model, was first introduced by Breiman [38] and has consistently shown its excellence in both classification and regression tasks in various domains [39]. Two RF strategies to secure the diversity of individual models in the ensemble population are (1) bootstrapping aggregating (bagging) and (2) randomly chosen variables for each split during tree growth. Once a training dataset is provided, different training sets called bootstraps are constructed by sampling with replacement. The number of instances in the bootstrap is the same as that of the original training dataset. By allowing replacement, the data distribution of bootstrapped training datasets is slightly distorted. For each bootstrapped dataset, a full decision tree is constructed. The main idea of RF is that when growing the tree, only randomly chosen variables can be used as the split variable. This variable selection process is repeated until the full tree is constructed. The number of candidate split variables is usually set to the square root of the original variable, as recommended in the original paper [38]. Adopting this recommendation, we set the number of candidate split variables to the square root of the number of input variables, and the number of bootstrapped training datasets was set to 500. Because the class imbalance is significant in our dataset, i.e., the good WBMs outnumber the bad WBMs, we modified the bootstrapping process to ensure balanced class distribution; equal number of good and bad wafers were sampled with replacement for each bootstrap. With this process, bad wafers have higher chances of being sampled than good wafers.

### 4.4. Performance Evaluation Criteria

Since the purpose of the classification model is to identify faulty wafers correctly, the confusion matrix can be summarized as Table 3. As performance measures, we used recall, precision, simple accuracy, balanced correction rate (BCR), and the F1-measure, all of which can be computed based on the confusion matrix.

**Table 3.** Confusion matrix for wafer bin map classification.

|  |  | Actual Class | |
|---|---|---|---|
|  |  | Bad (Positive) | Good (Negative) |
| **Predicted Class** | Bad (Positive) | True Positive (TP) | False Positive (FP) |
|  | Good (Negative) | False Negative (FN) | True Negative (TN) |

Recall is the ratio of the number of correctly classified bad wafers and the total number of bad wafers, whereas precision is the ratio of the number of correctly classified bad wafers and the total number of wafers classified as bad by the classification model, as shown in Equations (4) and (5), respectively:

$$Recall = \frac{TP}{TP + FN}, \tag{4}$$

$$Precision = \frac{TP}{TP + FP}. \tag{5}$$

A good classification model can achieve both high recall and precision. If recall is low, then many faulty wafers are missed by the model. If precision is low, then false alarms occur frequently. Simple accuracy is the ratio of correctly classified wafers regardless of the original class, as shown in Equation (6):

$$Accuracy = \frac{TP + TN}{TP + TN + FP + FN}. \tag{6}$$

Accuracy is not an appropriate performance measure when the class imbalance is significant, as in our case, since it yields high accuracy if the model classifies all test examples to the majority class, which is practically useless. Hence, BCR and the F1-measure could be more appropriate for an imbalanced dataset. BCR is the geometric mean of class-wise accuracy and the F1-measure is the harmonic mean of recall and precision, as shown in Equations (7) and (8), respectively:

$$BCR = \sqrt{\frac{TP}{TP + FN} \times \frac{TN}{TN + FP}}, \tag{7}$$

$$F1 - measure = \frac{2}{\frac{1}{Precision} + \frac{1}{TPR}}. \tag{8}$$

The five performance evaluation criteria described above depend on the cut-off values. To compare the intrinsic ability of classification models, cut-off independent performance evaluation criterion is needed. The receiver operating characteristic (ROC) curve shows the relationship between the false positive rate (*x*-axis) and true positive rate (*y*-axis) by varying the cut-off from tightest value to the loosest value. The ROC curve always starts from $(0, 0)$ and ends with $(1, 1)$. The area under the ROC curve (AUROC) is shown in Figure 14, and is commonly adopted as a cut-off independent performance metric. The AUROC ranges between 0.5 (random model) and 1 (ideal model); the larger the AUROC, the better the classification model.

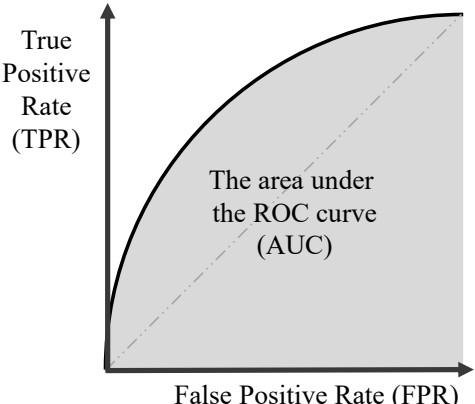

**Figure 14.** Receiver operating characteristic curve.

## 5. Results

In this study, we randomly split the entire dataset into 80% for training and the remaining 20% for testing. This process was repeated 30 times to secure more reliable results than a single experiment. Table 4 shows the classification performance of each model for the nine product groups and Figure 15 shows the box plots of the AUROC of three classification models for each product group. Bold numbers in the table are the highest values among the three classification models for each group.

**Table 4.** Classification of Good and Bad WBM results from group 1 to group 9 (Values without/with parenthesis are the average/standard deviation of 30 trials, respectively; in bold face are the best performance for each group and indices).

| Group | Model | Binary | | | | | | Random RGB | | | | | | Bin2Vec | | | | | |
|---|---|---|---|---|---|---|---|---|---|---|---|---|---|---|---|---|---|---|---|
| | | REC | PRE | ACC | F1 | BCR | AUC | REC | PRE | ACC | F1 | BCR | AUC | REC | PRE | ACC | F1 | BCR | AUC |
| 1 | MLP | 83.72 | 82.45 | 91.81 | 83.03 | 88.86 | 0.946 | 61.46 | 74.11 | 85.61 | 67.14 | 75.66 | 0.880 | 61.77 | 73.61 | 85.52 | 67.12 | 75.77 | 0.877 |
| | | (3.28) | (2.62) | (1.08) | (2.28) | (1.797) | (0.011) | (3.43) | (3.06) | (1.07) | (2.64) | (2.10) | (0.014) | (3.17) | (3.10) | (1.11) | (2.62) | (2.01) | (0.013) |
| | RF | 83.06 | 79.51 | 90.72 | 81.21 | 87.96 | 0.941 | 71.04 | 65.56 | 83.97 | 68.15 | 79.10 | 0.894 | 65.64 | 69.25 | 84.65 | 67.36 | 77.15 | 0.876 |
| | | (2.84) | (2.38) | (1.00) | (1.92) | (1.57) | (0.010) | (3.14) | (2.55) | (1.33) | (2.32) | (1.91) | (0.013) | (2.93) | (2.51) | (1.15) | (2.30) | (1.84) | (0.015) |
| | CNN | **91.40** | **92.85** | **96.25** | **92.10** | **94.53** | **0.989** | **88.56** | **91.04** | **95.17** | **89.76** | **92.80** | **0.981** | **87.06** | **91.04** | **94.84** | **88.98** | **92.02** | **0.971** |
| | | (1.97) | (1.90) | (0.65) | (1.37) | (1.04) | (0.004) | (2.04) | (1.85) | (0.66) | (1.40) | (1.09) | (0.005) | (2.10) | (1.95) | (0.55) | (1.19) | (1.02) | (0.006) |
| 2 | MLP | 89.07 | 88.93 | 97.25 | 88.95 | 93.60 | 0.972 | 89.53 | 90.06 | 97.45 | 89.74 | 93.92 | 0.972 | 86.40 | 89.77 | 97.07 | 88.02 | 92.28 | 0.970 |
| | | (3.82) | (2.41) | (0.55) | (2.33) | (2.01) | (0.016) | (4.04) | (3.12) | (0.67) | (2.75) | (2.14) | (0.016) | (3.43) | (2.40) | (0.57) | (2.43) | (1.88) | (0.019) |
| | RF | 87.21 | 84.17 | 96.37 | 85.63 | 92.28 | 0.975 | 87.10 | 84.57 | 96.43 | 85.78 | 92.25 | 0.978 | 86.97 | 83.36 | 96.23 | 85.08 | 92.09 | 0.976 |
| | | (3.39) | (2.51) | (0.61) | (2.34) | (1.83) | (0.009) | (3.24) | (2.24) | (0.53) | (2.07) | (1.72) | (0.008) | (3.51) | (1.90) | (0.46) | (1.94) | (1.83) | (0.008) |
| | CNN | **96.08** | **96.07** | **99.02** | **96.06** | **97.74** | **0.997** | **95.76** | **96.32** | **99.01** | **96.02** | **97.59** | **0.996** | **95.69** | **94.45** | **99.02** | **96.05** | **97.57** | **0.995** |
| | | (1.31) | (1.59) | (0.21) | (0.84) | (0.63) | (0.003) | (1.91) | (1.67) | (0.25) | (1.02) | (0.93) | (0.004) | (1.54) | (1.63) | (0.26) | (1.04) | (0.77) | (0.005) |
| 3 | MLP | 86.56 | 92.47 | 95.98 | 89.39 | 92.22 | 0.962 | 85.05 | 94.59 | 96.12 | 89.51 | 91.64 | 0.958 | 86.02 | 94.46 | 96.26 | 89.98 | 82.14 | 0.968 |
| | | (3.50) | (2.68) | (0.97) | (2.63) | (1.98) | (0.018) | (4.85) | (2.19) | (1.09) | (3.16) | (2.68) | (0.018) | (4.63) | (2.43) | (1.06) | (3.01) | (2.54) | (0.017) |
| | RF | 87.41 | 93.44 | 96.38 | 90.27 | 92.78 | 0.976 | 87.98 | 94.31 | 96.66 | 90.99 | 93.18 | 0.980 | 97.95 | 94.54 | 96.70 | 91.07 | 93.19 | 0.979 |
| | | (3.98) | (2.18) | (0.95) | (2.54) | (2.17) | (0.010) | (3.62) | (2.07) | (0.85) | (2.35) | (1.97) | (0.009) | (4.14) | (2.00) | (0.91) | (2.54) | (2.22) | (0.010) |
| | CNN | **96.15** | **97.69** | **98.80** | **96.90** | **97.78** | **0.992** | **95.91** | **98.36** | **98.88** | **97.11** | **97.74** | **0.992** | **95.99** | **98.12** | **98.85** | **97.03** | **97.75** | **0.991** |
| | | (1.33) | (1.28) | (0.30) | (0.77) | (0.64) | (0.005) | (1.31) | (1.08) | (0.32) | (0.83) | (0.67) | (0.005) | (1.56) | (0.89) | (0.35) | (0.92) | (0.80) | (0.005) |
| 4 | MLP | 62.11 | 70.40 | 95.27 | 65.81 | 77.86 | 0.897 | 63.74 | 74.26 | 95.67 | 68.30 | 78.99 | 0.911 | 60.24 | 73.00 | 95.39 | 65.70 | 76.77 | 0.892 |
| | | (6.98) | (5.95) | (0.69) | (5.59) | (4.48) | (0.036) | (6.95) | (5.95) | (0.63) | (5.04) | (4.36) | (0.032 ) | (7.17) | (6.73) | (0.70) | (5.49) | (4.56) | (0.030) |
| | RF | **85.07** | 55.00 | 93.83 | 66.67 | 89.61 | 0.944 | 83.59 | 58.37 | 94.42 | 68.55 | 89.19 | 0.948 | 84.23 | 58.52 | 94.48 | 68.93 | 89.53 | 0.949 |
| | | (5.21) | (4.89) | (0.90) | (4.27) | (2.67) | (0.019) | (5.36) | (6.66) | (1.07) | (5.53) | (2.89) | (0.018) | (5.21) | (6.21) | (1.03) | (5.47) | (2.83) | (0.018) |
| | CNN | 83.25 | **90.96** | **98.15** | **86.80** | **90.89** | **0.970** | **86.59** | **91.00** | **98.37** | **88.64** | **92.70** | **0.970** | **87.15** | **91.79** | **98.46** | **89.28** | **93.02** | **0.979** |
| | | (5.39) | (3.82) | (0.43) | (3.25) | (2.91) | (0.018) | (4.52) | (4.57) | (0.50) | (3.43) | (2.44) | (0.018) | (4.88) | (4.50) | (0.47) | (3.23) | (2.58) | (0.014) |
| 5 | MLP | 83.46 | 58.65 | 88.05 | 68.74 | 86.07 | 0.914 | 78.00 | 57.52 | 87.43 | 66.09 | 83.34 | 0.886 | 77.42 | 61.85 | 88.90 | 68.67 | 83.90 | 0.917 |
| | | (5.94) | (4.25) | (1.74) | (3.82) | (3.02) | (0.025) | (5.83) | (3.66) | (1.49) | (3.65) | (3.09) | (0.030) | (5.27) | (3.76) | (1.34) | (3.62) | (2.88) | (0.021) |
| | RF | 80.64 | 54.08 | 86.25 | 64.61 | 83.85 | 0.898 | 74.99 | 57.90 | 87.56 | 65.22 | 82.05 | 0.909 | 83.82 | 54.56 | 86.54 | 65.99 | 85.39 | 0.914 |
| | | (4.62) | (4.01) | (1.33) | (3.30) | (2.26) | (0.016) | (5.31) | (4.13) | (1.28) | (3.61) | (2.93) | (0.013) | (4.09) | (4.20) | (1.48) | (3.55) | (2.22) | (0.012) |
| | CNN | **85.17** | **89.87** | **96.10** | **87.27** | **91.38** | **0.986** | **90.04** | **89.52** | **96.75** | **89.71** | **93.92** | **0.984** | **89.50** | **89.93** | **96.75** | **89.63** | **93.68** | **0.986** |
| | | (4.77) | (4.45) | (0.77) | (2.40) | (2.26) | (0.007) | (2.85) | (3.74) | (0.68) | (2.04) | (1.40) | (0.009) | (3.62) | (3.37) | (0.64) | (2.04) | (1.79) | (0.010) |
| 6 | MLP | 47.30 | 84.23 | 58.48 | 51.99 | 49.77 | 0.301 | 92.59 | 91.70 | 90.58 | 92.13 | 90.05 | 0.974 | 92.27 | 91.81 | 90.45 | 92.00 | 89.96 | 0.975 |
| | | (32.57) | (12.88) | (5.90) | (17.30) | (2.71) | (0.017) | (2.09) | (1.40) | (1.67) | (1.43) | (1.66) | (0.006) | (2.85) | (1.58) | (1.47) | (1.34) | (1.38) | (0.006) |
| | RF | 28.40 | **89.56** | 54.94 | 41.64 | **49.88** | **0.321** | 93.18 | 90.81 | 90.35 | 91.95 | 89.65 | 0.967 | 93.01 | 90.78 | 90.25 | 91.86 | 89.56 | 0.970 |
| | | (12.73) | **(6.00)** | (2.51) | (7.20) | **(2.43)** | **(0.020)** | (2.17) | (1.88) | (1.49) | (1.28) | (1.55) | (0.009) | (2.33) | (1.94) | (1.62) | (1.41) | (1.66) | (0.008) |
| | CNN | **97.19** | 65.76 | **68.14** | **78.44** | 49.44 | 0.292 | **94.48** | **91.81** | **91.66** | **93.11** | **90.92** | **0.979** | **94.38** | **91.84** | **91.64** | **93.09** | **90.91** | **0.979** |
| | | (0.94) | (1.07) | (1.52) | (0.90) | (3.30) | (0.016) | (1.12) | (1.49) | (1.15) | (0.93) | (1.34) | (0.005) | (1.17) | (1.47) | (1.18) | (0.96) | (1.35) | (0.005) |
| 7 | MLP | 82.43 | 85.65 | 91.07 | 83.93 | 88.22 | 0.937 | 84.99 | 90.50 | 93.20 | 87.58 | 90.50 | 0.954 | 84.60 | 89.75 | 92.88 | 87.02 | 90.15 | 0.950 |
| | | (4.20) | (3.64) | (1.54) | (2.87) | (2.28) | (0.020) | (4.61) | (3.25) | (1.64) | (3.15) | (2.57) | (0.019) | (5.13) | (3.39) | (1.84) | (3.55) | (2.90) | (0.022) |
| | RF | 80.99 | 90.62 | 92.13 | 85.46 | 88.44 | 0.953 | 84.10 | 93.28 | 93.69 | 88.39 | 90.55 | 0.966 | 84.06 | 92.48 | 93.44 | 88.00 | 90.38 | 0.967 |
| | | (3.18) | (3.53) | (1.18) | (2.24) | (1.70) | (0.012) | (3.75) | (2.63) | (1.26) | (2.29) | (1.98) | (0.011) | (3.64) | (2.71) | (1.23) | (2.08) | (1.86) | (0.010) |
| | CNN | **94.60** | **96.67** | **97.53** | **95.60** | **96.62** | **0.992** | **95.47** | **97.04** | **97.88** | **96.23** | **97.13** | **0.992** | **95.35** | **96.71** | **97.76** | **96.01** | **97.01** | **0.990** |
| | | (1.97) | (1.95) | (0.75) | (1.34) | (1.04) | (0.005) | (1.63) | (1.62) | (0.59) | (1.04) | (0.83) | (0.004) | (1.76) | (1.68) | (0.63) | (1.12) | (0.89) | (0.004) |
| 8 | MLP | 67.55 | 61.51 | 88.91 | 64.21 | 79.00 | 0.858 | 68.81 | 79.56 | 92.78 | 73.66 | 81.59 | 0.886 | 66.16 | 84.97 | 93.26 | 74.25 | 80.44 | 0.884 |
| | | (6.43) | (5.80) | (1.71) | (5.02) | (3.79) | (0.038) | (5.73) | (4.89) | (1.14) | (4.35) | (3.39) | (0.040) | (5.08) | (5.29) | (0.98) | (3.90) | (3.03) | (0.033) |
| | RF | 63.74 | 84.14 | 93.11 | 72.36 | 78.96 | 0.878 | 71.90 | 85.09 | 94.17 | 77.66 | 83.81 | 0.956 | 70.84 | 87.28 | 94.34 | 77.00 | 83.37 | 0.944 |
| | | (5.68) | (4.21) | (1.18) | (3.93) | (3.46) | (0.048) | (5.45) | (5.45) | (1.16) | (4.20) | (3.83) | (0.020) | (5.82) | (4.88) | (1.13) | (3.81) | (3.36) | (0.020) |
| | CNN | **87.23** | **93.30** | **97.18** | **90.08** | **92.85** | **0.982** | **91.89** | **97.21** | **98.41** | **94.41** | **95.61** | **0.991** | **92.14** | **96.73** | **98.37** | **94.32** | **95.70** | **0.991** |
| | | (4.37) | (3.91) | (0.87) | (3.04) | (2.34) | (0.010) | (4.00) | (2.59) | (0.70) | (2.48) | (2.09) | (0.007) | (3.58) | (2.83) | (0.65) | (2.27) | (1.85) | (0.008) |
| 9 | MLP | 63.71 | 71.14 | 93.61 | 66.86 | 78.47 | 0.909 | 51.14 | 72.92 | 93.05 | 59.89 | 70.61 | 0.907 | 65.33 | 73.61 | 94.03 | 68.87 | 79.58 | 0.914 |
| | | (7.28) | (6.67) | (0.88) | (4.89) | (4.43) | (0.022) | (5.83) | (7.60) | (0.88) | (5.37) | (4.07) | (0.031) | (7.72) | (5.90) | (0.80) | (4.70) | (4.54) | (0.030) |
| | RF | 74.94 | 68.04 | 93.98 | 71.05 | 84.76 | 0.946 | 85.68 | 64.68 | 93.92 | 73.54 | 90.08 | 0.951 | 85.45 | 63.28 | 93.63 | 72.50 | 89.81 | 0.951 |
| | | (7.34) | (7.68) | (1.17) | (5.95) | (4.15) | (0.020) | (6.28) | (6.44) | (1.18) | (5.48) | (3.39) | (0.022) | (6.47) | (6.06) | (1.05) | (4.92) | (3.34) | (0.022) |
| | CNN | **85.24** | **89.05** | **97.40** | **86.95** | **91.72** | **0.989** | **87.81** | **90.62** | **97.81** | **89.04** | **93.16** | **0.990** | **86.67** | **90.58** | **97.69** | **88.43** | **92.57** | **0.989** |
| | | (5.31) | (4.98) | (0.70) | (3.61) | (2.86) | (0.010) | (5.56) | (4.01) | (0.62) | (3.24) | (2.92) | (0.010) | (4.82) | (4.83) | (0.61) | (3.02) | (2.49) | (0.012) |

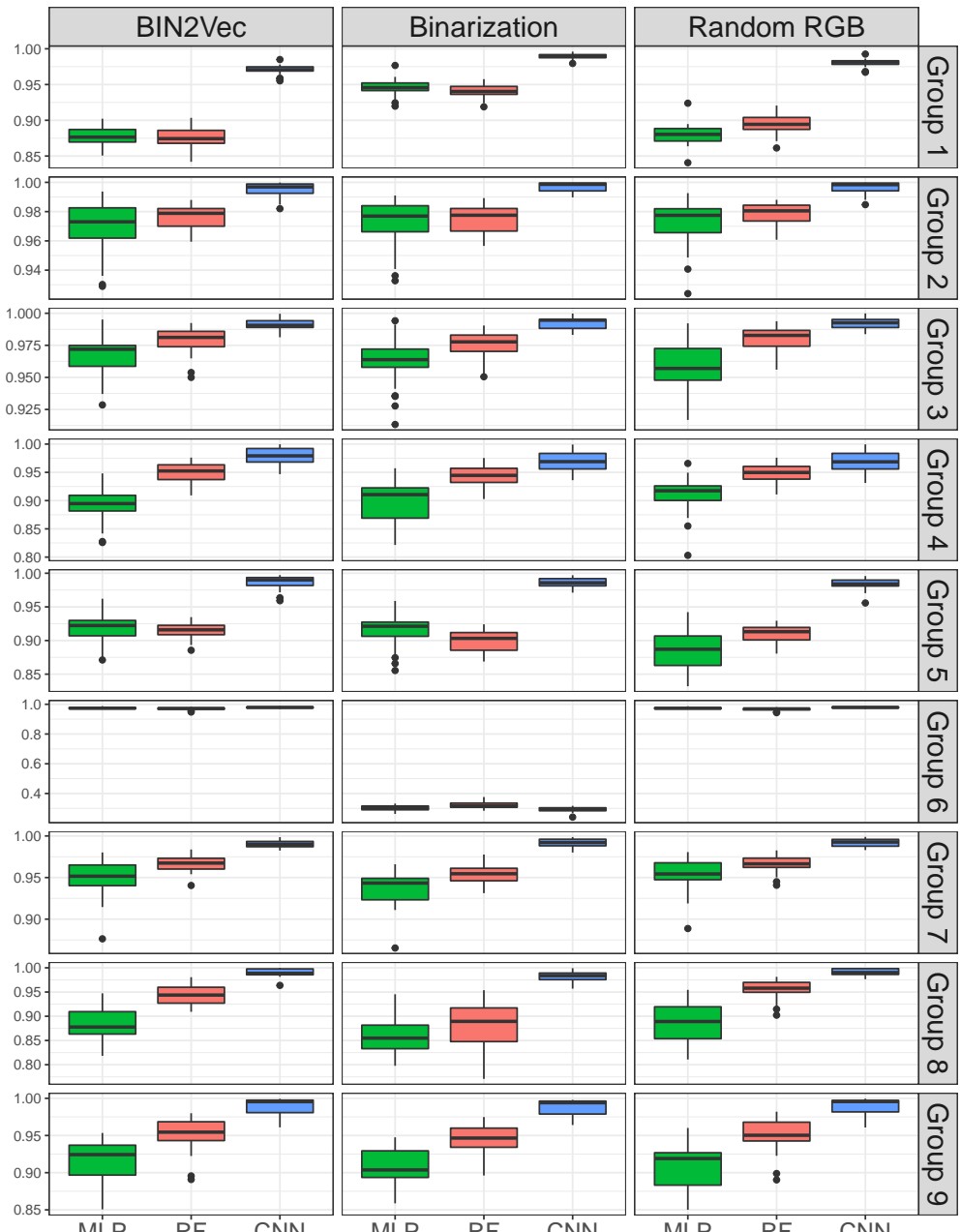

**Figure 15.** Box plot of DNN, RF, CNN, AUC results based on 1∼9 groups.

Except for product Group 6, which will be discussed later, the proposed CNN model outperformed the MLP and RF with all WBM coloring schemes for all performance evaluation criteria. Not only did it yield the best performance for all product groups, it also resulted in the most stable performance; the standard deviations for 30 repetitions are much smaller than those of MLP and RF. The BCR of the CNN model improved by 8.13% on average (min: −0.44% for product Group 6 with binary, max: 22.6% for product Group 9 with random RGB), and the F1-measure of the CNN model improved by 15.14% on average (min: 0.98% for Group 6 with random RGB, max: 36.8% for product Group 6 with binary). In addition, the AUC of the CNN model improved by 0.04 on average (min: −0.029 for product Group 6 with binary, max: 0.12 for Group 8 with binary). These improvements are all statistically significant at significance level of 0.01. Between the MLP and RF, RF generally yielded better bad wafer classification performance than MLP. The average differences between BCR, F1-measure, and AUC are 3.24%, 0.72%, and 0.02, respectively.

　　With respect to the WBM coloring scheme, both random RGB and Bin2Vec generally worked well for bad wafer classification. Their AUROC values are greater than 0.99 for all product groups. There are no significant differences between the random RGB and Bin2Vec in terms of bad wafer classification performance. In other words, although the bin codes on the wafer are represented by randomly assigned colors, different bin codes are distinguishable such that the bad wafer classification model can learn the difference between bad and good wafers with high accuracy. This implies that although the relationship between bin codes was are not configured, the weights of convolution operations can be appropriately learned for the purpose of accurate bad wafer classification.

　　An interesting observation is that when binarization was used to represent the WBM color, its bad wafer classification performance significantly decreased for product Group 6 irrespective of classification models. Its AUROC is 0.292 although that of random RGB and Bin2Vec for the same product group is 0.979. It can be concluded that for a certain wafer group, it is not sufficient to know whether each die passes all EDS tests or not, but additional information, i.e., which combination of EDS tests fails, must be provided for an improved classification performance.

　　The experimental results show that both random RGB and Bin2Vec provided satisfactory performance in the bad wafer classification task. However, Bin2Vec has the advantage of more in-depth analysis for discovering significant patterns among bad WBMs. Figure 16 shows all bad WBMs in a two-dimensional space reduced by the *t*-SNE method [40]. The output values of the second fully connected layer in Figure 12 were used as the input of the *t*-SNE algorithm. Based on Figure 16, we can identify some representative bad wafer patterns. For example, WBMs where most bins are beige are clustered at the top center area, next to where WBMs with green ring-shaped bins are located. WBMs with blue cross pattern are clustered at the bottom left area, and WBMs with green scattered bins are located near to the WBMs with blue cross pattern.

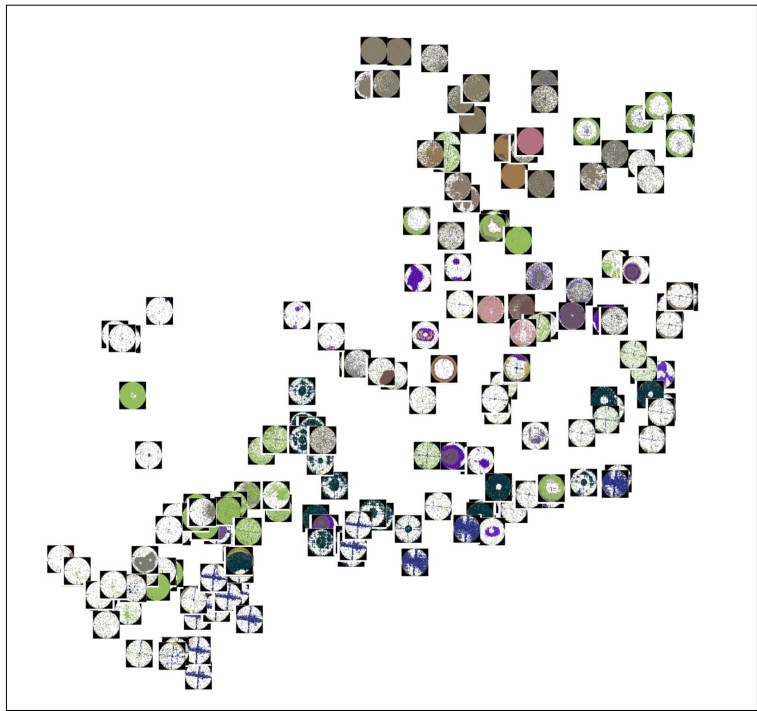

**Figure 16.** Visualizing bad wafers based on t-SNE and Bin2Vec.

　　Figure 17 shows the *t*-SNE results of the same wafers with the random RGB coloring scheme. In contrast to the Bin2Vec, it is difficult to discover semantic patterns from this figure. For example, not only the cross pattern in the bottom-right area but also various bad patterns are color green. In addition, some very different colors are mixed in single WBMs at the top-left area, which might

be confusing for engineers investigating bad bin patterns. Based on these two figures, it is more advantageous to use the Bin2Vec coloring scheme than random RGB because its color similarity within WBMs is more consistent with human perception than that of the random RGB coloring scheme.

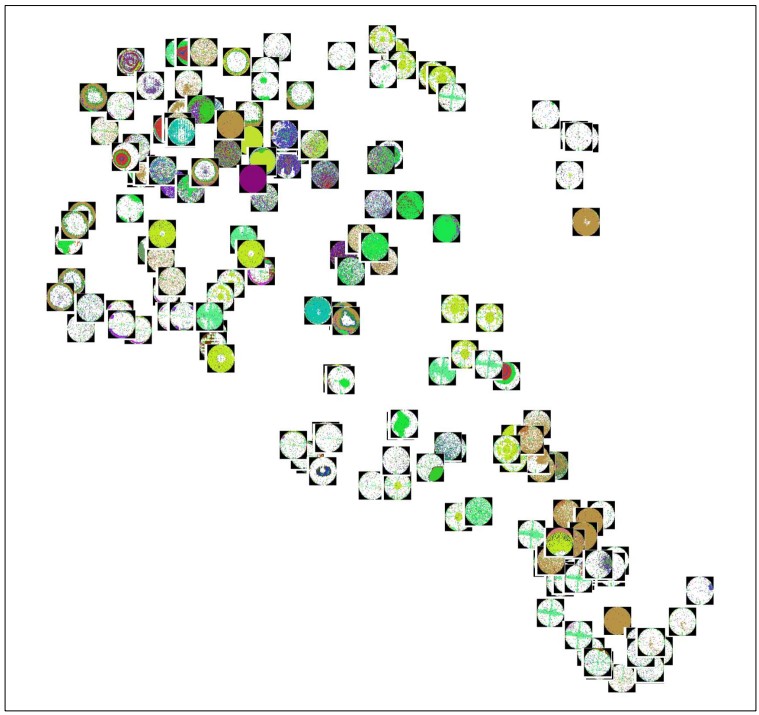

**Figure 17.** Visualizing bad wafers based on t-SNE and random RGB.

## 6. Conclusions

In this paper, we proposed a new, neural network-based WBM coloring scheme called Bin2Vec to preserve the relationship between different bin codes to better understand WBMs and find significant bad wafer patterns. Bin2Vec maps a scalar bin code onto a three-dimensional continuous vector in order to assign a unique set of RGB values to the bin code. In addition, we also built a CNN-based WBM classification model to automate the bad wafer classification process, which is done manually by engineers. The experimental results showed that it is easier to discover significant WBM patterns with Bin2Vec than with the random RGB method. In addition, the proposed CNN model with Bin2Vec yielded high classification accuracy regardless of the product type.

Despite the favorable experimental results, there are some limitations in the current study, which lead us to future research directions. First, although it was confirmed that Bin2Vec helped identify meaningful bad wafer patterns, it can be more efficient that the significant bad WBM patterns are automatically detected and grouped. This, in turn, raises the following questions: (1) Is this WBM good or bad? (2) If it is bad, to what type of WBM pattern does it belong? It is possible for WBM patterns to change over time. Therefore, it can be more helpful if a continuous WBM monitoring system is constructed.

**Author Contributions:** J.K. initiated the research idea and carried out the experiment. He also wrote the draft of the paper. H.K., J.P. and K.M. support the experiment. P.K. wrote and finalized the paper.

**Funding:** This research was supported by Basic Science Research Pro-gram through the National Research Foundation of Korea (NRF) funded by the Ministry of Education (NRF-2016R1D1A1B03930729) and Institute for Information & Communications Technology Promotion (IITP) grant funded by the Korea government (MSIP) (No. 2017-0-00349, Development of Media Streaming system with Machine Learning using QoE (Quality of Experience)). This work was also supported by Korea Electric Power Corporation (Grant number: R18XA05).

**Conflicts of Interest:** The authors declare no conflict of interest.

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
