# Peer review of "Bin2Vec: A Better Wafer Bin Map Coloring Scheme for Comprehensible Visualization and Effective Bad Wafer Classification"

_applsci, doi:10.3390/app9030597_

Round 1
Reviewer 1 Report
Please see the attached review report.

Author Response
Please see the attached response letter

Reviewer 2 Report
This paper investigates the coloring scheme method of wafer bin map (WBM) for wafer inspection in semiconductor manufacturing. If electrical die sorting tests for each wafer are represented by a random RGB method, we cannot reveal the relationship between different wafer bin codes. The authors propose a convolutional neural network (CNN) model to classify the bad wafer detection results. They attempt to preserve the physical or sematic relationship between the bin codes and coloring patterns which can be utilized to identify significant patterns on bad wafers. They design different experiments and give performance evaluation criteria to show the advantage of their proposed color scheme strategy. In addition, the authors point out the limitations of their proposed method.
Generally speaking, the authors have clear motivations and sound presentation/organization for this paper. There is a confusing issue for this reviewer. In line 88 of page 3, the authors claim that they propose a CNN-based bad wafer detection model to create a more accurate automatic wafer classification model. In the later sections, there are similar statements. It is Not appropriate. Essentially, this CNN is used to classify and analyze the given wafer detection results as stated in line 7 of page 1. It focuses on classification and adopts the EDS tests results as inputs which are represented by WBM image data. This paper does not cover the topic of EDS tests or detections. Therefore, “detection” in the title of this paper should be changed to “classification”.
Some minor issues are listed below.
1. In line 238, “increase”-> “increases”; in line 442 of page 17, “show”->”shows”
2. The sentence in lines 472-474 of page 17is vague.
3. The following NN-related papers should be commented and cited:
[A] D. Yu and J. Li. Recent progresses in deep learning based acoustic models. IEEE/CAA Journal of Automatica Sinica, 4(3), 396-409, 2017
[B] S. Gao, M. Zhou, Y. Wang, J. Cheng, H. Yachi, and J. Wang, "Dendritic neuron model with effective learning algorithms for classification, approximation and prediction," IEEE Transactions on Neural Networks and Learning Systems, 30(2), pp. 601 - 614, Feb. 2019.
[C] X.Feng, X. Kong, and H. Ma, “Coupled Cross-correlation Neural Network Algorithm for Principal Singular Triplet Extraction of a Cross-covariance Matrix,” IEEE/CAA Journal of Automatica Sinica, 3(2), 147-156, 2016
Author Response
Please see the attached response letter

Round 2
Reviewer 1 Report
The authors have addressed all issues arisen from the reviewer.
This manuscript is a resubmission of an earlier submission. The following is a list of the peer review reports and author responses from that submission.